REGISTERED REPORT PROTOCOL

# A public health community health worker-delivered intervention to reduce human trafficking among Denotified Tribes in India: A protocol paper

Meredith Dank[1*☯], Sheldon Zhang[2☯], Stephen Abeyta[1☯], Lauren Moton[1☯], Hanni Stoklosa[3☯], Nani Cuadrado[4☯], Alex Dahlen[1☯], David J. Farabee[5☯]

1 New York University, Marron Institute, New York, New York, United States of America, 2 University of Massachusetts, Lowell, Dept of Criminal Justice, Lowell, Massachusetts, United States of America, 3 Department of Medicine, Brigham and Women's Hospital, Boston, Massachusetts, United States of America, 4 HEAL Trafficking, Boston, Massachusetts, United States of America, 5 New York University Grossman School of Medicine, New York, New York, United States of America

☯ These authors contributed equally to this work.
* mdank@nyu.edu

This is a Registered Report and may have an associated publication; please check the article page on the journal site for any related articles.

## Abstract

The objective of this study is to evaluate an intervention designed to reduce human trafficking among Denotified Tribes (DNTs) in two regions of India. We will conduct a cluster-randomized controlled trial utilizing a participatory-designed, community health worker (CHW)- delivered public health intervention. CHWs will be trained to conduct anti-human-trafficking advocacy and psychological first aid (humane, supportive and practical assistance to people who are distressed) to DNT community members, and mobilize resources to ensure access to health and mental health services, education, livelihoods, and government benefits. This project leverages known effective, systemic, and sustainable approaches to reducing vulnerabilities to trafficking among DNT communities, through increased economic alternatives, health and mental health services guided by the trafficking-survivor-informed treatment protocols.

## Introduction

India has 1,500 nomadic and semi-nomadic tribes and 198 denotified tribes. These DNTs make up 10% of India's population [1]. Under the Criminal Tribes Act of 1871, these entire communities were stated to be criminal by nature and birth. Despite the repeal of the Act in 1949 and the "denotification" of former "criminal tribes" in 1952, the DNTs continue to experience stigma and systemic oppression. For example, eighty-nine percent of this population are landless and have little access to government-funded subsidies, which are tied to residence [2]. DNT members are deprived not only of access to social services, but they are also deprived of fundamental rights of citizenship.

For instance, government authorities require a Caste certificate before citizens can apply for health, education, or other social service assistance. Before such a certificate is granted, the applicant needs to provide proof of residence, the caste certificate of relatives, and a

**Data availability statement:** No datasets were generated or analysed during the current study. All relevant data from this study will be made available upon study completion.

**Funding:** This project was funded by the U.S. State Department (Grant # SSJTIP22CA0034). The opinions, findings and conclusions stated herein are those of the authors and do not necessarily reflect those of the United States Department of State.

**Competing interests:** The authors have declared that no competing interests exist.

birth certificate. These requirements are onerous for DNT members, who may not have the required documents and face disinterest from government workers [1,2]. To sustain their livelihood, DNTs migrate from state to state and eke out a living from seasonal farming, domestic work, acrobatics, dancing, snake charming, hunting, fortune-telling, brewing liquor, begging, handicrafts, and construction work. Low-skilled laborers find themselves victims of physical and verbal abuse, coercion and forced labor practices. Chronic poverty, illiteracy, unemployment, health complications, and substandard living conditions compound their vulnerability and can keep DNT families in subjugation. Additionally, of the 198 DNT communities in India, six to ten tribes are known to participate in intergenerational commercial sex where this work became the tribe's primary source of livelihood [1]. The combination of abject poverty and common participation in the commercial sex can create conditions in which DNT members become exploited often rising to the level of human trafficking. The term "human trafficking" in this study is defined by the US Trafficking Victims Protection Act of 2000 and operationalized by the US State Department's Office to Monitor and Combat Trafficking in Persons through its PRIF (Prevalence Reduction Innovation Forum) program where participation in the sex trade or abusive labor practices must meet specific thresholds to qualify as human trafficking violations [3]. Human trafficking among DNTs has been overlooked by anti-trafficking efforts and research. The DNT community identifies the following gaps in anti-trafficking response: lack of access to health care, government benefits, alternative livelihood pathways, and harm reduction tools. Lack of access to government benefits and alternative livelihood pathways means DNT members have limited options to support their families and are often exploited for their labor. Many lack skills, awareness, and opportunities to prevent force, fraud, or coercion in the workplace. Many factors drive these gaps, including stigma faced by DNTs, their lack of trust in external resources and ability to pursue alternative livelihood pathways, lack of political will, and a lack of DNT community advocates' ability to build community members' resiliency. Poverty and exploitation in the DNT communities is seemingly accepted, both by DNTs and broader Indian society, and attempts to change the social status of DNTs and their access to resources often fail. Although recent actions have tried to correct discrimination toward the DNT community, many DNTs are unaware of or unable to access their entitled benefits. One study documented DNT members refusing to access benefits, such as children's education, due to negative responses from school administrators, teachers, and other students [4]. India's DNT special commission, established in 2005, has had limited impact on the plight of DNTs. Society's general suspicion of, direct aggression towards, and absence of any legal protection for DNTs are ongoing. There is also a stigma against those engaged in commercial sex and victims of sex trafficking in India, reinforcing the perception of DNT members as "less than." The present study seeks to address these gaps by first reporting the baseline experiences of DNTs in two Indian districts, one in the state of Madhya Pradesh and the other in Tamil Nadu. This baseline study will then inform and create transformational change through the development of a Community Health Worker (CHW)-delivered public health, community-based participatory designed intervention.

## Materials and methods

### Intervention

The intervention seeks to address different DNT vulnerabilities to trafficking through psychological first aid and supported referrals that are guided by the trafficking-survivor-informed, evidence-based, Privacy Educate Ask Respect Respond (PEARR) Tool and (Health, Education, Advocacy, Linkage) HEAL Protocol Toolkit [5]. The PEAAR approach is to

enable participatory engagement through informative conversations with the participants that promote health, safety, and wellbeing in safe environments that encourage patients to open and accept further services. Although PEARR was developed by Western medical professionals, it is intended to be a vehicle to identify important community service needs and plan intervention activities through a co-creation process where the beneficiaries voice their needs and participate in a continuous feedback loop to improve the delivery. The project will hire and train two types of community health workers (CHWs): Community Mobilizers (CM) and Health Fellows (HF). The CMs will be responsible for community mobilization and ensuring access to education, livelihoods, and entitlements. HFs will be responsible for health access activities. CMs and HFs will be trained in psychological first aid counseling and trafficking. Our in-country partner has decades of field and intervention experience working with these communities, while the North America team provides technical support on impact assessment, monitoring and evaluation design, as well as administration of the funds. By participatory design, this study ensures all target communities take part in the design, planning, and implementation of the intervention; and all interventionists, as described below, are recruited directly from the communities. At the core of our causal pathway is a harm-reduction approach that uses the public health vehicle to advocate for the empowerment of the DNT communities to build collective efficacy in achieving greater access to government entitlement programs and more resources to develop alternative livelihoods. At the start, this project employs motivational interviewing and trauma-informed safety planning to help participants develop resilience and empower them to navigate life challenges and secure gainful employment free from exploitation. For our supported referrals, we will utilize the survivor informed and evidence based HEAL Protocol Toolkit that is used in over 35 countries to craft minimum standards for trafficking response in health care settings [6]. The CMs will facilitate participants' contact with trafficking victim service advocates who can provide legal advice, alternative incomes, vocational training access, and healthcare access. Within each treatment hamlet, a Community Awareness and Action Group (CAAG) will be formed and participatory mapping methods will be implemented to select community actions and review intervention performance. The goal is to connect community members to health care, counseling, education opportunities, government resources (such as caste-based entitlements), and alternative pathways to livelihood in order to reduce trafficking in these communities. To improve access to government-sponsored entitlements and social security programs, we will train CMs on key provisions of social security programs and application procedures. They will develop community awareness of programs and kiosks (a location within the hamlet where the CM will assist the community member with registering for government schemes) to assist members in submitting applications. Their program activities will be informed by the community level mapping of key issues and aspirations related to social security programs. The CHFs and CAAG will work together in all aspects of the program, review performances and adjust activities accordingly. Ongoing training and supervision by the research team will support CHFs and maximize their ability to engage and assist intervention participants.

## Impact evaluation

We will use a cluster-randomized trial (CRT) to evaluate the impact of the proposed intervention. We will use household- and individual- level survey measures to evaluate work exploitation and human trafficking risk factors. We will compare the hamlets that receive our intervention to a set of matched control hamlets, as follows.

**Pairing of eligible hamlets prior to randomized assignment.** Intervention and matched hamlets were chosen in a three-step process. First, DNT hamlets were screened for eligibility

and logistical feasibility. This resulted in 62 eligible hamlets in the project site in Tamil Nadu and 30 in the project site in Madhya Pradesh. Second, the eligible hamlets were clustered based on demographic and socio-economic factors that were measured in our initial community mapping exercise. Specifically, clusters were formed using the following features: the number of households, gender breakdown, number of children, caste certifications obtained, and general household income. Similarity scores were based on Gower similarity scores. We identified 10 clusters of hamlets in the project site in Tamil Nadu and 6 clusters in the project site in Madhya Pradesh. Third, pairs of hamlets within each cluster were identified and screened by our in-country partner to ensure that no pairs were physically adjacent to one another, to prevent possible crossover contaminations. Pairs were randomly selected for the study, and hamlets within each pair were randomly assigned to the intervention group or to control group. Table 1 shows the resulting pairs of hamlets in our study.

**Participant screening.**   We applied a household-based census approach to screen all eligible households in these selected hamlets. Based on census data (one site had 217 households in the treatment group and the other had 166) and community mapping, our field partner built an exhaustive list of all households in the selected hamlets. Upon arrival at a DNT hamlet, our team of enumerators used the list of all dwelling units to initiate the data collection process. Within each household, we identified a single eligible participant who was between the ages of 18-55 to complete the survey. If a household had more than one eligible participant, we applied the birthdate scheme to select whoever had the birthdate closest to the enumerator visit date. If all eligible respondents were absent at the time of the field visit but were still living in the household (i.e., had not migrated elsewhere), information would be sought on when the individual would return and up to two additional attempts were made. All households in the selected hamlets were contacted, and our recruitment procedure ensured all eligible members of each hamlet would have equal chances of being included in the study. The baseline survey (which has already occurred) was administered on a computer tablet and lasted anywhere between 45-60 minutes.

Table 1.  Randomized clusters of treatment and control hamlets.

| Treatment | Num. Households | Control | | Num.Households |
|---|---|---|---|---|
| **Madhya Pradesh Site:** | | | | |
| Tx_Hamlet1 | 15 | | | |
| Tx_Hamlet2 | 18 | | | |
| Tx_Hamlet3 | 6 | Ctrl_Hamlet1 | | 20 |
| Tx_Hamlet4 | 95 | Ctrl_Hamlet2 | | 19 |
| Tx_Hamlet5 | 26 | Ctrl_Hamlet3 | | 18 |
| Tx_Hamlet6 | 90 | Ctrl_Hamlet4 | | 120 |
| *Total households* | *250* | Ctrl_Hamlet5 | | 70 |
| **Tamil Nadu Site:** | | NA | | |
| Tx_Hamlet1 | 35 | | *Total households* | *247* |
| Tx_Hamlet2 | 39 | | | |
| Tx_Hamlet3 | 47 | Ctrl_Hamlet1 | | 41 |
| Tx_Hamlet4 | 37 | Ctrl_Hamlet2 | | 58 |
| Tx_Hamlet5 | 48 | Ctrl_Hamlet3 | | 39 |
| Tx_Hamlet6 | 15 | Ctrl_Hamlet4 | | 36 |
| *Total households* | *221* | | *Total households* | *232* |

*Hamlet names are anonymized at the request of local implementation partner due to political sensitivity.

Consent was obtained at beginning of the survey, with the following information explained to the participants:

1. The survey was voluntary, participant may stop at any time.

2. Participation or non-participation will not affect relationships with our field partner.

3. Information would be kept confidential. Will be assigned unique ID and responses would not be shared except in aggregate.

4. Participant may stop the interview at any time.

5. Enumerators would confirm that participant understood their rights and had no questions prior to the commencement of the survey interview.

6. Enumerators were to conduct the interviews in a space where conversation could not be heard by others.

Research questions and hypotheses The main research question in this project is whether a community-based participatory intervention will empower the DNT communities to increase economic opportunities and access to government entitlements, thus reducing the collective risk exposure to human trafficking. We seek to apply a clustered randomized trial (CRT) and a rigorous Monitoring/Evaluation/Research/Learning (MERL) process to ensure a high fidelity in implementation of the proposed intervention. Further, this project has the following hypotheses, that relative to the control hamlets, in reducing at a greater rate individual-level risk of trafficking or/and experiences of trafficking violations:

(1) HFs recruited from treatment DNT hamlets and trained to deliver a multi-pronged intervention, including psychological first aid, community health interventions in collaboration with the CAAGs.

(2) CMs recruited from treatment DNT hamlets and trained will empower their communities to access more entitlements and social security programs, through connecting community members in participating areas to health care resources, governmental resources, and alternative pathways to livelihood.

(3) Residents of the treatment hamlets will report having more alternative livelihood opportunities, and more success in entrepreneurship and economic innovations.

(4) Residents in the treatment hamlets will report having greater access to basic literacy and secondary education opportunities as a result of tutoring, transportation and application assistance.

(5) Fewer residents of the treatment hamlets will be victimized by workplace exploitations and trafficking violations.

## Study organization and sites

This project is a collaboration between New York University, HEAL Trafficking, and our local Indian partner agency. The study sites are hamlets located in the State of Madhya Pradesh and the State of Tamil Nadu, India. At the request of our local partner, we have decided not to name the specific study sites and our local implementation agency due to political sensitivity.

### Regulatory affairs and data and safety monitoring

This study protocol was reviewed and approved by the New York University Institutional Review Board. IRB-FY2023-7128. The protocol was registered with the Evidence in Governance and Politics (EGAP) registry (ID: 20230925AA). The study protocol was also reviewed and approved by the Ethics Review Board of Praxis, a participatory research institute, in India.

### Participants

In all, 790 households (one adult participant surveyed from each household) were enrolled in the evaluation, including 382 (48%) from the treatment hamlets and 408 (52%) from the control hamlets. Of these household heads, 44% were male. Most of these participants were between 26 and 40 years old. Twenty-five percent had received at least 9 years of formal education; 63% reported being married at the time of the survey, and 47% reported that they did not have a caste certificate.

### Assessments

To assess baseline rates and to evaluate our impact, we designed a survey instrument to quantify the prevalence of trafficking-prone activities and trafficking victimization. We also produced a short version of the survey which can be completed in 10-15 minutes for monitoring purposes. The longer survey will be administered at baseline and endline in all hamlets; the shorter, monitoring survey will be administered quarterly in the intervention hamlets.

We applied two strategies to establish baseline measurement to gauge changes in human trafficking in this population and intervention- outcome: (1) prevalence in trafficking-prone activities, namely employment and livelihoods, in these DNT hamlets, and (2) prevalence in trafficking victimization among these hamlets. The intent is to measure changes in both the extent of participation by DNT hamlets in trafficking- prone activities and their actual human-trafficking victimization. Once the baseline is established, we will repeat the survey at the endline to compare changes between the two clusters over time. A list of targeted outcomes for the project is shown in Table 2.Considering the fact that some of the guiding concepts in program design and implementation were developed with by Western medical professionals, as in the PEARR program, this study relies on the close involvement from our in-country partner to develop a detailed monitoring and evaluation (M&E) platform to assess and validate the cultural competency of our intervention activities. Program monitoring activities (e.g., activity tracking, pre/post tests, and observations) are entered monthly into the M&E system and reports are shared between the implementation partner in India and the technical support in North America. Our in-country partner maintain daily communication with their teams of CHWs and CMs in their communities to ensure timely feedback and corrective actions.

### Survey instrumentation

This project built an instrument that assesses the following domains from respondents: (1) demographic profiles of the respondent and his/her family members; (2) employment activities, (3) work and earnings, (4) exploitative practices, (5) rights violations, and (6) engagement with services, law enforcement. The surveys also included the Short Form-12 [7], along with measures of trafficking experiences [8,9], healthcare access [10], healthcare spending [11], and stages of change (adapted from the URICA [7]. A copy of the baseline survey is included as Supplementary Material. A copy of the baseline survey is included as Supplementary Material.

**Table 2. Outcome metrics (collected quarterly).**

| Category | Outcome | Source |
|---|---|---|
| Exploitation/Trafficking | | |
| Primary outcome | (1) % who experienced at least one type of workplace exploitation<br>(2) % who experienced enough workplace exploitations/abuses to be considered potential victims of human trafficking | Endline and quarterly surveys |
| Secondary outcomes | % of participants involved in other at-risk/illicit economic activities<br>% experiencing trafficking victimization or participating in trafficking- prone activities | Endline and quarterly surveys<br>Endline and quarterly surveys |
| Risk Factors | | |
| Employment type<br>Employment type<br>Schooling | % of respondents engaged in commercial sex in any capacity<br>% of participants who have or secure a living wage through a non-exploitative/illicit job during the project<br>% of children in the household going to school/ who have dropped out of school | Endline and quarterly surveys<br>Endline and quarterly surveys<br>Endline and quarterly surveys |
| Healthcare<br>Government | % who have seen a health care worker in their hamlet<br>(1) #/% of participants who have obtained or applied to social security or other government services | Endline and quarterly surveys<br>Endline and quarterly surveys |
| | (2) #/% of participants who have obtained or applied to entitlement documents<br>(3) % with a caste certificate<br>(4) % who worked under MGNREGA (Mahatma Ghandi National Rural Employment Guarantee Act) | |

## Treatment conditions

The study design involves random assignment of hamlets to an early or delayed intervention condition. This allows us to assess the impact of the intervention while ensuring that all randomized participants will eventually receive it.

## Statistical analysis plan

This evaluation will rely on a cluster-randomized trial occurring in 23 hamlets: 11 in a project site in the state of Madhya Pradesh in the north and 12 in a project site in the state of Tamil Nadu in the south of India. Several factors were considered for the site selections, including geographical representation, available hamlets of comparable sizes and sufficient distances from one another to allow randomized assignment and sufficient distance to minimize spillover effects, and access to the physical locations of the hamlets. Study participants in the intervention group will respond to a quarterly survey over the course of the study to assess receipt of the proposed intervention activities, and all study participants will participate in a more in-depth baseline and follow-up interview. The latter will allow for the direct assessment of the intervention's impact and will also provide a wide range of covariates. Based on the baseline survey, the percentages of those reporting exploitation (the primary outcome) in the preceding five years was similar between study conditions (Test hamlets: 57.3% ± 5.0% vs Control hamlets: 61.9% ± 4.7%).

## Analytic approach

Our primary statistical approach will be a matched difference-in-differences comparisons of the rates of work exploitation between test and control hamlets. We will compare the average

change between the two timepoints in the test hamlets to the control hamlets. In addition to overall exploitation, we will consider the secondary outcomes in Table 2. In addition, we will analyze the quarterly monitoring survey using a longitudinal, multi-level mixed-effects logistic model of the form logit (E(exploitation)) = $\Sigma\beta \times$ Time Point + $\lambda i + \eta j$, where $\lambda i$ is a person-level random intercept to account for the multiple time periods, and $\eta j$ is a hamlet-level random intercept to account for clustering. For both analysis methods, we will consider a test for heterogenous treatment effects by site (Madhya Pradesh v. Tamil Nadu) using inter-action terms. *Missing data*. The following measures were taken to minimize missing data: If a study participant is not reached the first time, the field staff will follow up at least one more time, including by texting/calling. If a study participant still cannot be reached, the field staff will ask someone else in the household to fill it out for them. The research team will regu-larly check the missingness levels for each question. If any question has a high rate of being skipped, the team will investigate and consider rewriting the question.

Despite these strategies, some missing data is inevitable. Our primary strategy will be to use MICE with 10 dataset replications, using Rubin's rules to pool point estimates and standard errors. As a sensitivity analysis, we will also report: 1) the complete case result; 2) the worst-case result, where missing values are imputed as y=1 indicating exploitation; 3) the best-case result, where missing values are imputed as y=0, indicating no exploitation.

## Discussion

Historical discrimination against members of DNTs has resulted in limited access to criti-cal services, including healthcare, education, and basic government services like obtaining documentation or land titles. As a result, DNT members are at greater risk of maltreatment, exploitation and trafficking. Using an intent-to-treat cluster randomized trial, this project will assess the effectiveness of an intervention designed to address these service gaps for this popu-lation. The results of the final evaluation are expected in fall of 2027.There are several limita-tions in this study. First, our intervention activities focus on two geographical regions in India, thus limiting this study's generalizability to other DNT communities across India. Secondly, it is important to point out, but difficult to fully mitigate, the risk of a Hawthorne effect in this project. We should note that both treatment and control hamlets have experienced increased attention from the research team and our collaborating partner as part of the initial mapping, surveying, intervention development, and baseline interviews.

## Limitations

Despite the use of a randomized design and established survey measures, this evaluation has at least three important limitations. First, the outcomes measures rely on self-reports, which can be problematic when assessing sensitive behaviors). Second, given that the intervention may raise respondents' awareness of exploitation, it is possible that participa-tion could increase *reported* levels of exploitation, even if such events remained stable. One approach to address the latter limitation will be to focus on exploitative conditions and *events*, rather than self-reported perceptions of the crime. Lastly, due to the lack of liveli-hood opportunities in and around the treatment hamlets, outward household migration (i.e., case attrition) may become an issue during the intervention period. We may apply statistical procedures to impute outcomes for lost participants, although we assume that the attrition rates will be similar across both treatment and control hamlets due to similar broader socio-economic environments. In addition, we will conduct a sensitivity analysis comparing outcomes using the imputed values versus outcomes obtained by treating val-ues from missing participants as missing.

## Acknowledgements

We are grateful for our local staff based in Tamil Nadu and Madhya Pradesh for enumerating the households in the participating hamlets and conducting the surveys for this project. We are also grateful to the residents of the participating hamlets for providing input in the development of the intervention.

## Author contributions

**Conceptualization:** Sheldon Zhang, David J. Farabee.

**Data curation:** Sheldon Zhang, Stephen Abeyta, Alex Dahlen, David J. Farabee.

**Formal analysis:** Sheldon Zhang, Stephen Abeyta, Alex Dahlen, David J. Farabee.

**Funding acquisition:** Meredith Dank, Sheldon Zhang.

**Methodology:** Sheldon Zhang, David J. Farabee.

**Project administration:** Meredith Dank, Lauren Moton.

**Supervision:** Meredith Dank.

**Validation:** Sheldon Zhang, Alex Dahlen, David J. Farabee.

**Visualization:** Sheldon Zhang, David J. Farabee.

**Writing – original draft:** Meredith Dank, Sheldon Zhang, Stephen Abeyta, Alex Dahlen, David J. Farabee.

**Writing – review & editing:** Meredith Dank, Sheldon Zhang, Stephen Abeyta, Lauren Moton, Hanni Stoklosa, Nani Cuadrado, Alex Dahlen, David J. Farabee.

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
