## [Decision Letter · Decision Letter 0]

8 Nov 2024

PONE-D-24-33545A Public Health Community Health Worker-Delivered Intervention to Reduce Human Trafficking Among Denotified Tribes in India: A Protocol PaperPLOS ONE

Dear Dr. Dank,

Thank you for submitting your manuscript to PLOS ONE. After careful consideration, we feel that it has merit but does not fully meet PLOS ONE’s publication criteria as it currently stands. Therefore, we invite you to submit a revised version of the manuscript that addresses the points raised during the review process.

We look forward to receiving your revised manuscript.

Kind regards,

Kasi Eswarappa

Academic Editor

PLOS ONE

**Journal Requirements:**

2. In your cover letter, please confirm that the research you have described in your manuscript, including participant recruitment, data collection, modification, or processing, has not started and will not start until after your paper has been accepted to the journal (assuming data need to be collected or participants recruited specifically for your study). In order to proceed with your submission, you must provide confirmation.

3. Please include a complete copy of PLOS’ questionnaire on inclusivity in global research in your revised manuscript. Our policy for research in this area aims to improve transparency in the reporting of research performed outside of researchers’ own country or community. The policy applies to researchers who have travelled to a different country to conduct research, research with Indigenous populations or their lands, and research on cultural artefacts. The questionnaire can also be requested at the journal’s discretion for any other submissions, even if these conditions are not met.  Please find more information on the policy and a link to download a blank copy of the questionnaire here: https://journals.plos.org/plosone/s/best-practices-in-research-reporting. Please upload a completed version of your questionnaire as Supporting Information when you resubmit your manuscript.

This project was funded by the U.S. State Department (Grant # SSJTIP22CA0034). The opinions, findings and conclusions stated herein are those of the authors and do not necessarily reflect those of the United States Department of State. 

This project was funded by the U.S. State Department (Grant # SSJTIP22CA0034). The opinions, findings and conclusions stated herein are those of the authors and do not necessarily reflect those of the United States Department of State. We are grateful for our local staff based in Tamil Nadu and Madhya Pradesh for enumerating the households in the participating hamlets and conducting the surveys for this project. We are also grateful to the residents of the participating hamlets for providing input in the development of the intervention. 

This project was funded by the U.S. State Department (Grant # SSJTIP22CA0034). The opinions, findings and conclusions stated herein are those of the authors and do not necessarily reflect those of the United States Department of State. 

**Additional Editor Comments:**

Reviewers advised to do minor revisions of the paper. Thus, I suggest you to do minor revisions of the manuscript.

***Comments from Editorial Office: ** Please note that the comments of Reviewer #2 are attached as a separate document, please refer to this during your revisions.*

Reviewers' comments:

Reviewer's Responses to Questions

**Comments to the Author**

1. Does the manuscript provide a valid rationale for the proposed study, with clearly identified and justified research questions?

Reviewer #1: Yes

Reviewer #2: Yes

2. Is the protocol technically sound and planned in a manner that will lead to a meaningful outcome and allow testing the stated hypotheses?

Reviewer #1: Partly

Reviewer #2: Yes

3. Is the methodology feasible and described in sufficient detail to allow the work to be replicable?

Reviewer #1: No

Reviewer #2: Yes

4. Have the authors described where all data underlying the findings will be made available when the study is complete?

Reviewer #1: Yes

Reviewer #2: Yes

5. Is the manuscript presented in an intelligible fashion and written in standard English?

Reviewer #1: Yes

Reviewer #2: Yes

6. Review Comments to the Author

You may also provide optional suggestions and comments to authors that they might find helpful in planning their study.

**Reviewer #1:**  Thank you very much for your work on this project, which will benefit a population in need of services, as well as their respective communities. The project is admirable, and I congratulate you on your ambitions and the local professional connections that make the project possible.

To clarify the protocol, I recommend several considerations, with less important recommendations toward the end of the list:

- The protocol refers broadly to the term "human trafficking," but that term has different interpretations in different nations and contexts. Please specify inclusion/exclusion descriptions and procedures. For instance: What about extended family members who willingly engage in sex work or physical labor that would appear from an American perspective to be exploitative, but the local norms and legal codes accept such arrangements?

- This study was reviewed by an IRB in the USA, but given the vast cultural differences, a review by an equivalent entity in India would be best practice, although not required.

- The word "participatory" appears throughout the manuscript, but the researchers appear to be interfacing with the community health workers and other interventionists, rather than the members of the denotified tribes being served. Since the term "participatory research" refers to those benefitting from services, rather than to interventionists from a different background, please replace all instances of the word “participatory” with more accurate terms and descriptions, such as "with the collaboration of local service providers." (This study was apparently designed in the USA, rather than by members of denotified tribes).

- The intervention was developed in an entirely different context than the two regions of India where the study will occur. So clarify that the intervention was not developed in India for the population targeted by this study. Although an intervention developed in Europe or North America may be commonly exported for use in other countries, that does not provide evidence of validity with the local population in this study. Local context matters, with cross-cultural research optimally including steps of cultural adaptation and validation (e.g., member checks in pilot tests to confirm that the interventions are received as intended). In this protocol, please list any cultural adaptations planned/made for the denotified tribes (e.g., given how they experience fellow Indians as outsiders).

- Since the intervention is described broadly, please specify how the intervention targets the specific outcomes evaluated. If there is an explanation for causality based on extant theory, that would be helpful to add but not essential.

- Please specify any fidelity checks of the community health workers’ training and interventions.

- Please specify if any individuals from denotified tribes serve as cultural brokers, interfacing with the community health workers. Specify if all study personnel working with the individuals from denotified tribes are native or fluent in the same primary language - and specify the procedures for working with individuals with limited abilities to read.

- Please focus the manuscript on the two regions/tribes involved in this study, rather than implying in the text that the project has relevance to all denotified tribes.

- For the design, be aware that attention by community health workers will “in and of itself” modify community expectations and behavior (Hawthorne effect), so possibly consider ways to differentiate the effects of the intervention from the effects of attention and visits from outsiders.

Again, this is a superb project that made me feel grateful for and proud of your work. I have offered suggestions for strengthening the protocol, but the most important fact is that you have boldly designed a meaningful contribution to our world. If projects like this occurred with greater frequency, imagine the impact. Thank you, thank you.

**Reviewer #2:**  Dear Editor,

I have added my comments in the Word document that is attached to this email. I was occupied with other tasks, which caused a one-day delay in the submission. I apologise for the inconvenience and appreciate your patience and understanding.

Thanking you,

Best regards,

Hemangi Kadlak, PhD

7. PLOS authors have the option to publish the peer review history of their article (what does this mean? ). If published, this will include your full peer review and any attached files.

**Do you want your identity to be public for this peer review?** For information about this choice, including consent withdrawal, please see our Privacy Policy .

Reviewer #1: No

Reviewer #2: **Yes: ** Hemangi Kadlak

---

## [Author Response · Author response to Decision Letter 0]

2 Dec 2024

November 20, 2024

Kasi Eswarappa

Academic Editor

PLOS ONE

RE: PONE-D-24-33545

Dear Dr. Eswarappa:

Thank you for sending the comments regarding to our manuscript “A Public Health Community Health Worker-Delivered Intervention to Reduce Human Trafficking Among Denotified Tribes in India: A Protocol Paper.” We were pleased with the positive reviews and have revised the manuscript to address the reviewers’ concerns. These revisions are summarized in italics below, and we have uploaded both marked and clean copies of the revised manuscript.

Reviewer #1

• The protocol refers broadly to the term "human trafficking," but that term has different interpretations in different nations and contexts. Please specify inclusion/exclusion descriptions and procedures. For instance: What about extended family members who willingly engage in sex work or physical labor that would appear from an American perspective to be exploitative, but the local norms and legal codes accept such arrangements?

Response: Thank you for raising this important definitional issue. Because of our funding source, this study adopts the term “human trafficking” as legally defined by the US Trafficking Victims Protection Act of 2000 and operationalized by the US State Department’s Office to Monitor and Combat Trafficking in Persons through its PRIF (Prevalence Reduction Innovation Forum) program. As suggested, we have added texts to clarify the use of this terminology.

• This study was reviewed by an IRB in the USA, but given the vast cultural differences, a review by an equivalent entity in India would be best practice, although not required.

Response: Thank you for pointing out this omission. The study was reviewed and approved by an in-country research institute’s IRB. We have added this information under section “Regulatory affairs and data and safety monitoring”.

• The word "participatory" appears throughout the manuscript, but the researchers appear to be interfacing with the community health workers and other interventionists, rather than the members of the denotified tribes being served. Since the term "participatory research" refers to those benefitting from services, rather than to interventionists from a different background, please replace all instances of the word “participatory” with more accurate terms and descriptions, such as "with the collaboration of local service providers."

Response: Thank you for pointing out this apparent lack of clarity in our description of the personnel hired to implement the program. To be clear, all interventionists, i.e., community health workers (CHWs) and community mobilizers (CMs), are recruited directly from the targeted DNT communities. They are trained and deployed back to the communities where they came from. We have added texts to clarify this point, at the end of the Introduction section.

• The intervention was developed in an entirely different context than the two regions of India where the study will occur. So clarify that the intervention was not developed in India for the population targeted by this study. Although an intervention developed in Europe or North America may be commonly exported for use in other countries, that does not provide evidence of validity with the local population in this study. Local context matters, with cross-cultural research optimally including steps of cultural adaptation and validation (e.g., member checks in pilot tests to confirm that the interventions are received as intended). In this protocol, please list any cultural adaptations planned/made for the denotified tribes (e.g., given how they experience fellow Indians as outsiders).

Response: Thank you for bringing up the cross-cultural aspect to our intervention planning and implementation. This is an important point, and we have added additional texts to clarify the participatory nature of our intervention design and implementation and efforts to assess the cultural adaption of our intervention strategy, in the first paragraph under the Intervention section, and in the final paragraph under the Assessments section. Although developed in North America, the PEARR program relies heavily on a participatory process in the design and implementation of its interventions. Our in-country partner brings forward decades of field experiences and intervention know-how from working with these communities, while the North America team provides technical support in impact assessment, monitoring and evaluation design, as well as administration of the funds. Because of the current political situation where the Indian government suspects the intentions of Western-funded NGO programs, our in-country partner has requested to remove the agency name as well as the names of all participating communities to avoid unwanted attention from the authorities.

• Since the intervention is described broadly, please specify how the intervention targets the specific outcomes evaluated. If there is an explanation for causality based on extant theory, that would be helpful to add but not essential.

Response: due to space limitation, we have refrained from expanding our theory of change, which centers around a harm-reduction orientation through a public health vehicle and seeks to gain trust in DNT communities that have grown suspicious to outside agencies. Through a participatory process, we seek to empower DNT communities to increase access to government entitlement programs, improve awareness of health and rights protection, and develop alternative livelihoods. We have added additional texts to clarify this causal pathway in the second paragraph under the Intervention section.

• Please specify any fidelity checks of the community health workers’ training and interventions.

Response: Thank you for pointing out this important monitoring task. All community health workers who receive training by our trainers will be assessed through several mechanisms to ensure fidelity: (1) assessment during classroom training activities; (2) pre/post tests at the onset and conclusion of the training; 3) close oversight of our in-country collaborating agency during course of the project; and (4) our established M&E feedback loop. We provided all intervention personnel standardized forms for them to log their activities (number of people served, types of services delivered, etc.). These forms will be monitored by both the local collaborating agency as well as the NYU project director. We have added this information in the final paragraph under the Assessment section.

• Please specify if any individuals from denotified tribes serve as cultural brokers, interfacing with the community health workers. Specify if all study personnel working with the individuals from denotified tribes are native or fluent in the same primary language - and specify the procedures for working with individuals with limited abilities to read.

Response: this is addressed in our response to comment #3 above. All intervention personnel are members of the DNT communities. The whole intension is to build collective agency that can sustain over time.

• Please focus the manuscript on the two regions/tribes involved in this study, rather than implying in the text that the project has relevance to all denotified tribes.

Response: We have added this important caveat to the limitations section of the Discussion.

• For the design, be aware that attention by community health workers will “in and of itself” modify community expectations and behavior (Hawthorne effect), so possibly consider ways to differentiate the effects of the intervention from the effects of attention and visits from outsiders.

Response: This is an important point. It is difficult to fully mitigate the risk of a Hawthorne effect in this project, but we should note that both treatment and control hamlets have experienced increased attention from the research team and our collaborating partner as part of the initial mapping, surveying, intervention development, and baseline interviews. We have added this problem—and our attempt to address it—to the Discussion.

Reviewer 2

• It will be good if you list out the tribes’ names.

Response: Thank you for the inquiry. There are well known tribes that engage in commercial sex trade because of their marginalized economic status, such as Bedia, Nat, Perna, Banchara, Rajnat, Bawaria, Kanjar, etc. However, to avoid further stigmatization, we opted not to list their tribe names in any of our publications. Doing so serves no particular purposes other than publicizing which tribes we may be working with, as their general geographical locations are known to the public. For political reasons, we have also omitted specific locations where our study is taking place in this manuscript.

• Reasons to choose these two states - Madhya Pradesh and Tamil Nadu.

Response: Thank you for the inquiry. We have added clarifications of the factors that went into the site selection in the text, with track change on. Briefly, our study intends to achieve some geographical representation because there are distinct differences between DNT tribes in the north and south of India. Further, we had to consider hamlets with comparable sizes in population and also sufficient distances from one another to allow randomization and to minimize potential cross-contamination. Finally, we had to consider access to the physical locations of these communities.

We are grateful for these comments and believe the revised manuscript is much-improved as a result. Please let us know if you have any further questions or comments.

Best,

Meredith Dank, PhD

** Funder information has been removed and highlighting has been removed as well.

---

## [Editor Report · Decision Letter 1]

5 Jan 2025

A Public Health Community Health Worker-Delivered Intervention to Reduce Human Trafficking Among Denotified Tribes in India: A Protocol Paper

PONE-D-24-33545R1

Dear Dr. Dank,

We’re pleased to inform you that your manuscript has been judged scientifically suitable for publication and will be formally accepted for publication once it meets all outstanding technical requirements.

Kind regards,

Kasi Eswarappa

Academic Editor

PLOS ONE

Additional Editor Comments (optional):

We are satisfied with your minor revisions and accepting it for publication.
---

## [Editor Report · Acceptance letter]

PONE-D-24-33545R1

PLOS ONE

Dear Dr. Dank,

I'm pleased to inform you that your manuscript has been deemed suitable for publication in PLOS ONE. Congratulations! Your manuscript is now being handed over to our production team.

Kind regards,

on behalf of

Dr. Kasi Eswarappa

Academic Editor

PLOS ONE